# Stereotactic Radiotherapy Boost in Treatment of Persistent Periocular Sebaceous Carcinoma after Surgery

**DOI:** 10.3390/biomedicines11061538

**Published:** 2023-05-26

**Authors:** Paweł Polanowski, Aleksandra Nasiek, Aleksandra Grządziel, Ewa Chmielik, Agnieszka Pietruszka, Krzysztof Składowski, Katarzyna Polanowska

**Affiliations:** 11st Radiation and Clinical Oncology Department, Maria Sklodowska-Curie National Research Institute of Oncology, Gliwice Branch, Wybrzeże Armii Krajowej 15, 44-102 Gliwice, Poland; pawel.polanowski@io.gliwice.pl (P.P.); krzysztof.skladowski@io.gliwice.pl (K.S.); 23rd Radiation and Clinical Oncology Department, Maria Sklodowska-Curie National Research Institute of Oncology, Gliwice Branch, Wybrzeże Armii Krajowej 15, 44-102 Gliwice, Poland; 3Radiotherapy Planning Department, Maria Sklodowska-Curie National Research Institute of Oncology, Gliwice Branch, Wybrzeże Armii Krajowej 15, 44-102 Gliwice, Poland; aleksandra.grzadziel@io.gliwice.pl; 4Tumor Pathology Department, Maria Sklodowska-Curie National Research Institute of Oncology, Gliwice Branch, Wybrzeże Armii Krajowej 15, 44-102 Gliwice, Poland; ewa.chmielik@io.gliwice.pl; 5Department of Clinical Oncology, Maria Sklodowska-Curie National Research Institute of Oncology, Cracow Branch, Garncarska 11, 31-115 Cracow, Poland; agnieszka.pietruszka@onkologia.krakow.pl; 6Ophthalmology Department, St. Barbara Provincial Hospital No 5, Plac Medyków 1, 41-200 Sosnowiec, Poland; polanowskakatarzyna@gmail.com

**Keywords:** sebaceous carcinoma, stereotactic radiotherapy, radiosurgery boost

## Abstract

Sebaceous carcinoma is a rare malignancy that should be treated with surgical resection. Nonetheless, a dynamic and aggressive course of the disease may disqualify a patient from this treatment. Applying radiotherapy with the escalation dose using a stereotactic boost is worthy of consideration as a radical treatment. In this paper, we present the case study of a young patient with a tumor localized in the periocular area. The patient was treated with operation two times without a satisfactory effect. Conventional radiotherapy, 60 Gy in 30 fractions, combined with chemotherapy based on cisplatin 40 mg/m^2^ and the addition of a stereotactic radiosurgery boost were administered. The tolerance of this treatment was acceptable. During the 2-year follow-up, local and distant recurrences were not diagnosed. The presented case shows the usefulness of an individualized approach in the radical treatment of sebaceous carcinoma with the use of the stereotactic radiotherapy boost. This is a subsequent example of the implementation of the boost in head and neck carcinoma, which yields a positive result.

## 1. Introduction

Sebaceous carcinoma is a rare and aggressive malignancy that accounts for less than 1% of skin cancer. The skin of the head and neck regions, especially the periocular area (upper eyelid), is predisposed to the localization of this tumor. It is derived from sebaceous glands, usually in patients over 60 years of age. Female sex, immunosuppression, previous irradiation, a genetic predisposition to the Muir–Torre syndrome, and familial retinoblastoma are confirmed risk factors. Radical operation is fundamental in clinical management. Radiotherapy is recommended as an adjuvant treatment with positive postsurgical margins or as a primary treatment in cases of unresectable tumors. The recurrence of sebaceous carcinoma as a local tumor (12%) or regional lymph node involvement (7%) demands an individual approach using surgery or radiotherapy. There is no effective systemic treatment. In metastatic disease, systemic therapy, including platinum-based or anthracycline-based regiments or immunotherapy (pembrolizumab), can be considered [1,2]. The presented case report shows difficulties and challenges in the therapeutic process and the usefulness of the stereotactic radiotherapy boost in the radical treatment of the dynamic course of sebaceous carcinoma.

## 2. Case Presentation

A 22-year-old white female patient without comorbidities and addictions presented with periodic headaches and right eye tearing for 6 months. A tumor in the upper-medial site of the right orbit appeared 2 months before a medical consultation. A histopathological examination based on surgical biopsy, which had been previously conducted, was ambiguous and there was a need to differentiate between sebaceoma and sebaceous carcinoma. Taking into account cellular atypia, necrosis, the elevated mitotic index, moreover positive expression of p53 and the focal positivity androgen receptors (AR) in neoplastic cells, the diagnosis of sebaceous carcinoma was finally signed. A computed tomography (CT) scan of the head and neck revealed a contrast-enhanced lesion measuring 30 × 30 mm, infiltrating the right nasal and lacrimal bone, the ethmoid bone and the frontal process of the right maxilla, growing into the nasal cavity. Additionally, magnetic resonance imaging (MRI) showed a contrast-enhanced tumor with restricted diffusion, encompassing the skin of the upper eyelid and adhering to the right medial rectus muscle. Positron emission tomography-computed tomography (18F-FDG PET-CT) confirmed a sole lesion with the maximum standardized uptake value of (SUVmax) 5.46 without distant metastases. The patient was qualified for surgery with a fibular free flap reconstruction. The surgeon verified the tumor relative to the right eyeball intraoperatively and decided to not remove the eyeball due to a lack of evidence of the infiltration. Lymphadenectomy was not performed. Histopathological examination showed sebaceous carcinoma G2 at stage IV (pT4b cN0 cM0, AJCC 8th edition), R1 resection. Additional risk factors included angioinvasion and Ki 67 20%. Afterward, during the case conference, the medical team ordered MRI to assess surgery effects before qualifying the patient for radiotherapy. It revealed a contrast-enhanced tumor with restricted diffusion measuring 16 × 14 mm in the upper-lateral part of the tumor bed infiltrating the right eyeball. Due to tumor persistence, the medical team decided to resign from radiotherapy and qualified the patient for reoperation—orbital exenteration with anterolateral thigh flap reconstruction. There were 111 days between the first and second operations. The histopathological examination demonstrated again sebaceous carcinoma G2 at Stage IV (rpT4b cN0 M0, AJCC 8th edition), R1 resection. Unfortunately, 48 days after reoperation, an MRI scan of the head and neck revealed residual infiltration with restricted diffusion in the tumor bed measuring 15 × 10 mm, again. The patient was disqualified from surgery and qualified for conventional radiotherapy with a stereotactic boost. A five-point head, neck and shoulder mask with 7 mm thermoplastic bolus was fitted, and a CT scan (3 mm slice thickness) without intravenous contrast and MRI with gadolinium intravenous contrast (1 mm slice thickness) in the supine position were performed to precisely identify the gross tumor volume (GTV). The dose prescription on the clinical target volume (CTV) involved the tumor bed with residual infiltration, nasal root, right orbit, nasal cavity, ethmoid bone, right maxillary sinus, right pterygopalatine fossa and right zygomatic bone to a total dose of 50 Gy in 25 fractions (CTV1) and residual infiltration with 5 mm margin to a total dose of 60 Gy in 30 fractions (CTV2). A 3 mm margin was added to the CTV creating the planning target volume (PTV). The VMAT (volumetric modulated arc therapy) technique was applied. In the first stage of conventional RT, two arcs limited to the right side of the patient were utilized; the first one with the range of gantry rotation from 180° to 30° and 0° table rotation, and the second one with gantry rotation from 30° to 330° and table position of 90°. A similar arrangement of two arcs was used in the second stage of treatment, i.e., gantry rotation from 341° to 30° with a table in 0° position and gantry rotation from 30° to 330° with a table position of 90°.The dose analysis of conventional radiotherapy is presented in Table 1 and Figure 1.

Radiotherapy was performed using a linear accelerator (Clinac 23EX; Varian Medical Systems, Palo Alto, CA, USA) with a nominal photon energy of 6 MV and a maximal beam rate of 600 MU/min.

A new mask with a 7 mm thermoplastic bolus, as well as a CT scan and MRI, were performed on the day of the twenty-third fraction of conventional treatment. GTV was 1.6 cm^3^ and was the same in comparison to GTV measured before the beginning of conventional treatment. A 3 mm margin was added to the GTV, creating PTV. The multidisciplinary case conference decided to apply a single-fraction stereotactic radiosurgery (SRS) boost of 12 Gy to the residual tumor. The VMAT technique was applied again. The setup of the two arcs was as follows: one from 220° to 46° with a table position of 0°, and the second one from 30° to 330° with a table rotation equal to 0°. Flattening Filter Free 6MVphoton beams with a maximal beam rate of 1400 MU/min were used. The dose analysis of the stereotactic radiotherapy is presented in Table 2 and Figure 2.

In planning the stereotactic boost, we gained the minimal dose of 11.80 Gy, the maximal dose of 13.09 Gy and the mean dose of 12.44 Gy in PTV. The boost was applied 4 days after the last fraction of conventional treatment. Concurrently, five cycles of 40 mg/m^2^ cisplatin were administered weekly during the conventional treatment. Dermatitis of the forehead and mucositis of the throat, both in stage G2 (CTCAE v.4.0), were observed as early side effects from the third week of radiotherapy. In the first follow-up visit, 3 months after the end of the treatment, a complete regression of the primary tumor was observed. Swelling of the right side of the face, a 5 mm bulge in the scar after the operation and skin redness of the forehead were also found during the physical examination. MRI was performed after another 2 months. It revealed a 6 mm contrast-enhanced zone, corresponding to the bulge mentioned above, and fluid in the maxillary and frontal sinuses on the right side, implying post-radiation changes. The subsequent follow-up visits were performed every 4–5 months; they did not reveal a recurrence of cancer for 24 months. Changes in sinuses on the right side were presented during the duration of the follow-up. The only symptom that was occasionally reported by the patient was blepharitis of the left eye.

Magnetic resonance imaging before, during and after the therapeutic process are presented below (Figure 3, Figure 4, Figure 5 and Figure 6).

## 3. Discussion

Periocular sebaceous carcinoma occurs rarely. Surgery based on complete circumferential peripheral and deep margin assessment, Mohs micrographic surgery or wide local excision are principal methods of treatment. R0 resection is the main goal and due to unfavorable localization on the face, it may require exenteration [1,2,3]. Dutch researchers described a group of 100 patients with sebaceous carcinoma of a different localization after surgical resection. The recurrence was confirmed in 17 cases and it was significantly higher in periocular tumors, as well as with positive postoperative margins [4]. Our patient was operated on twice—the tumor was removed the first time and the second operation was the exenteration needed due to residual infiltration. Histopathological reports indicated R1 resection in either case, and risk factors indicated an aggressive course of the disease. Data from the literature concerning radiotherapy and systemic therapy are poor. A retrospective review of 1349 patients with sebaceous carcinoma pointed out the rarity of using radiotherapy because only 5.4% of patients received it [5]. The authors of this paper did not provide precise indications for qualification for radiotherapy. McGrath et al. presented methods of management of recurrent sebaceous gland carcinoma after surgery. Of the 62 patients, 10 (16%) cases were diagnosed with recurrent disease. Most of them were treated with the operation, cryotherapy or topical interferon alpha-2a. The exenteration by external beam radiotherapy that followed was performed in only one patient. Perhaps, the reason for the lack of qualification for radiotherapy was the historical acknowledgment of sebaceous carcinoma as a radioresistant tumor. Solid evidence on the role of radiotherapy has provided a retrospective analysis of 83 patients treated for sebaceous carcinoma of the eyelid in Japan [6]. Radiotherapy had curative intent in all cases with the median total dose of 60 Gy—in 78% of patients initially as definitive radiotherapy and in 22% as postoperative radiotherapy. The 7-year overall survival and local control rates were 83.5% and 52.3%, respectively. Researchers proved that tumor size ≤10 mm was related to higher 7-year local control and freedom from neck lymph node recurrence. Late toxicity of radiotherapy was positive because only one patient revealed eyelid dysfunction of Grade 3. Beneficial data from the employment of radiotherapy in the adjuvant or definitive treatment for sebaceous carcinoma of the eyelid were presented by Hata [7]. Thirteen elder patients were irradiated to a total dose of 50–66.6 Gy (median, 60 Gy) in 22–37 fractions. During a 5-year observation time, the overall local progression–free and disease-free rates were 100% and 89%, respectively, and toxicity did not exceed Grade 2. Only one patient after an operation and adjuvant radiotherapy had a nodal recurrence. These data are examples of the utility of radiotherapy in the therapeutic process of sebaceous carcinoma. The role of systemic treatment is not well-documented. Kibbi et al. report in their review article individual cases treated with adriamycin, cyclophosphamide or 5-fluorouracil. In advanced cases, carboplatin or cisplatin was used as a neoadjuvant treatment for salvaging the globe. Moreover, pembrolizumab concomitant with carboplatin was an effective treatment in diffuse metastases and brought about a complete response [8]. Being aware of the unfavorable features of the disease in our young patient, we decided to escalate radiotherapy with the application of the stereotactic boost as the only chance for long-standing survival. The idea of combining conventional radiotherapy with stereotactic radiosurgery stems from an attempt to increase the biological effective dose (BED) in the target volume and is possible with using unconventional methods of fractionation. One of the clinical trials conducted at the National Research Institute of Oncology in Poland aimed to evaluate the efficacy and safety of a stereotactic boost applied in a range from 10 Gy to 18 Gy [9]. Although the protocol of clinical trial permits using even 18 Gy in tumors <7 cm^3^, the decision to apply a 12 Gy dose of stereotactic boost was made based on the periocular localization and the close distance to organs at risk. For the sake of calculations, a α/β ratio = 10 Gy was assumed. The combined biological effective dose was 98.4 Gy (72 Gy from conventional radiotherapy and 26.4 Gy from the stereotactic radiotherapy boost). Our previous experiences, which considered using a stereotactic radiotherapy boost on patients with adenoid cystic carcinoma and squamous cell carcinoma confirm the usefulness of such an approach in the radioresistant or dynamic course of cancer [9,10]. Early and late toxicities in those cases and in the case of the presented patient were acceptable; however, the application of a stereotactic radiotherapy boost needs very careful qualification and further clinical studies. The addition of cisplatin during conventional radiotherapy was supposed to sensitize the patient to radiation and decrease the tumor, but the GTV volume was the same after radiochemotherapy as it was in the beginning of the treatment. A similar modality was published by American authors who wrote about a group of 34 patients with locally advanced oropharyngeal cancer. Concurrent radiochemotherapy with a total dose of 60–66 Gy was followed by a radiosurgery boost on the residual tumor one week after conventional radiotherapy was administered in a single fraction of 8 Gy or 10 Gy or two fractions of 5 Gy each. Locoregional and distant control after a median follow-up of 50 months was 85.3% and 88.2%, respectively, without significant difference between different doses of the boost. The most serious adverse event was hemorrhage in Grade 4 requiring surgical intervention, which developed in three patients [11]. The forgoing examples clearly show the high effectiveness of stereotactic boosts despite the fact that it can be associated with potentially life-threatening toxicity.

## 4. Conclusions

The stereotactic radiosurgery boost is a noteworthy method of oncological treatment, especially in unresectable tumors. The implementation of the boost to radical schemes of radiotherapy or radiochemotherapy could be effective in dealing with intractable (refractory) cases. Due to the lack of strong data on the radiosurgery dose, its application should be deliberate and careful. It seems that, other than the optimistic results regarding the effectiveness of combining the stereotactic radiosurgery boost with conventional treatment, tolerance to this treatment will also have crucial meaning.

## Figures and Tables

**Figure 1 biomedicines-11-01538-f001:**
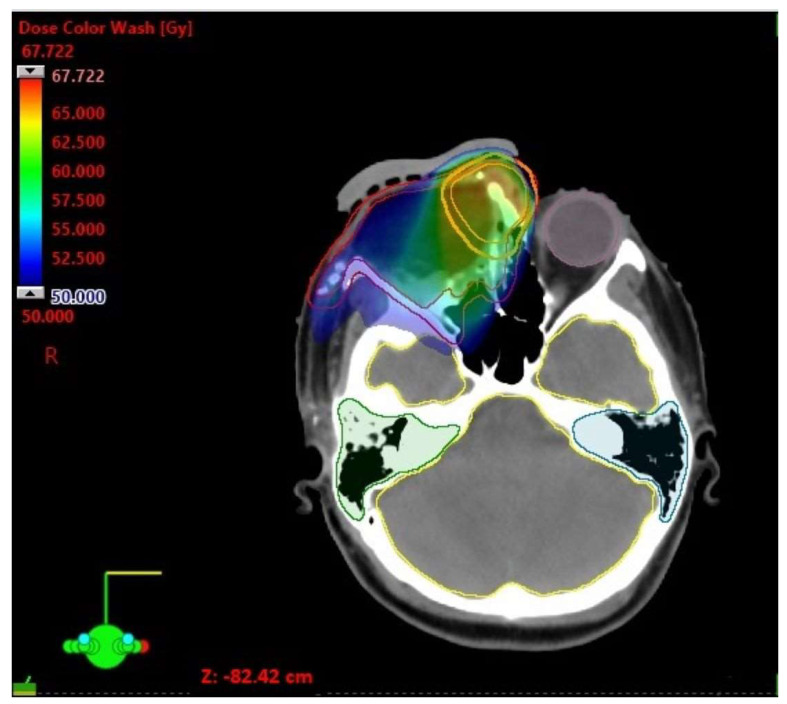
Conventional radiotherapy. Dose distribution shown on the CT scan.

**Figure 2 biomedicines-11-01538-f002:**
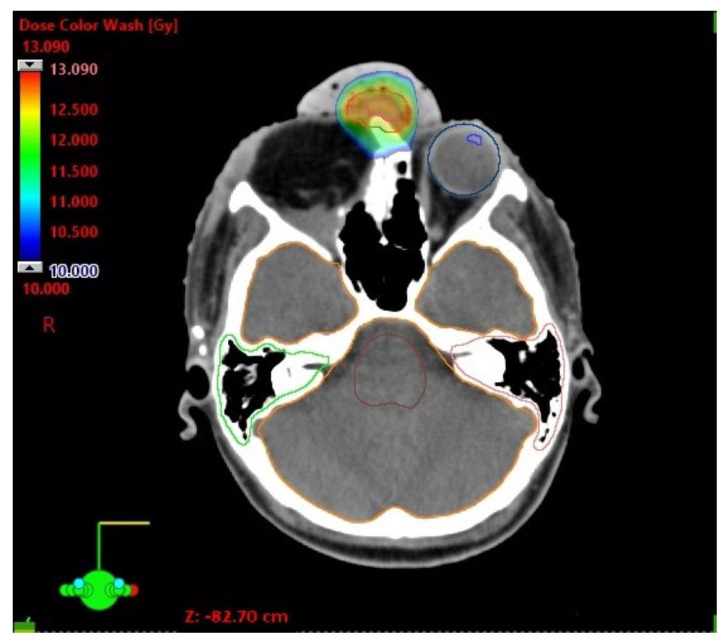
Stereotactic radiotherapy. Dose distribution shown on the CT scan.

**Figure 3 biomedicines-11-01538-f003:**
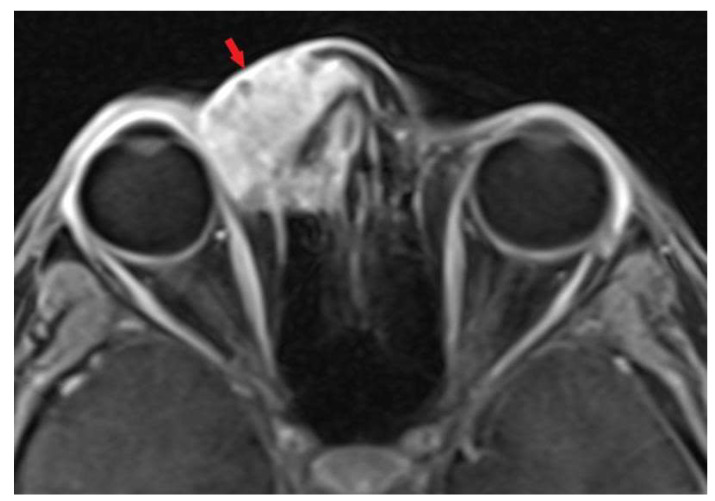
MRI before the treatment (t1 vibe). Red arrow indicates tumor infiltration.

**Figure 4 biomedicines-11-01538-f004:**
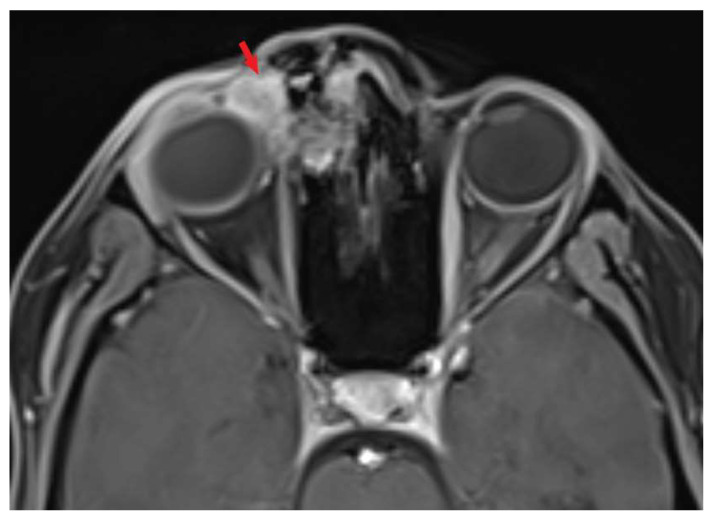
MRI after the first operation (t1 vibe). Red arrow indicates tumor infiltration.

**Figure 5 biomedicines-11-01538-f005:**
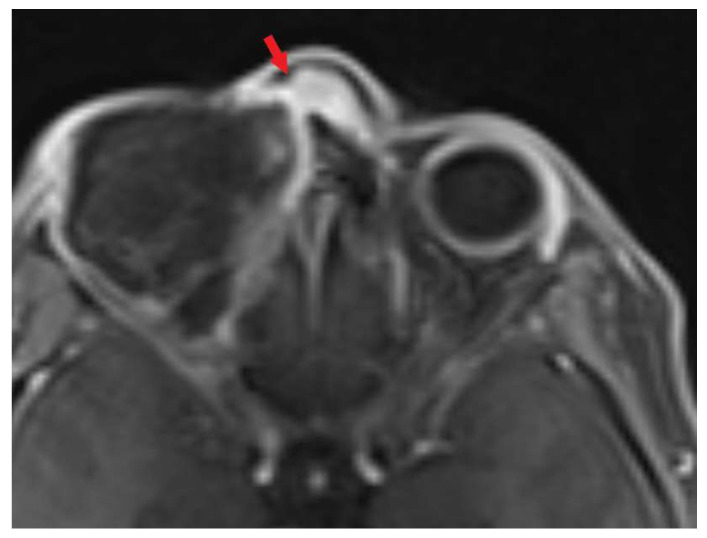
MRI after the second operation (t1 vibe). Red arrow indicates tumor infiltration.

**Figure 6 biomedicines-11-01538-f006:**
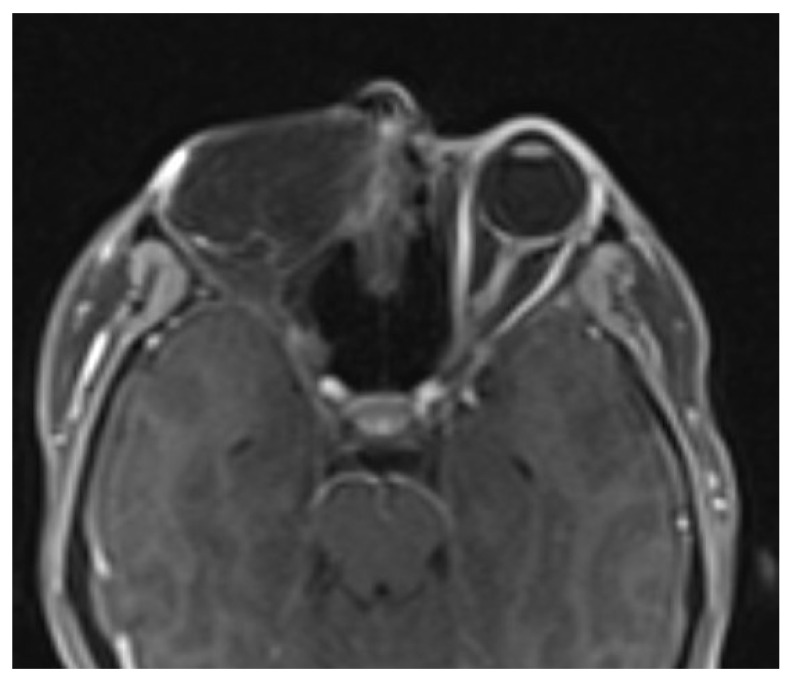
MRI three months after radiochemotherapy + stereotactic boost.

**Table 1 biomedicines-11-01538-t001:** Conventional radiotherapy—dose analysis.

	D Min [Gy]	D Max [Gy]	D Mean [Gy]
CTV 50	36.25	67.72	55.68
PTV 50	6.76	67.72	54.12
CTV 60	57.24	67.72	62.84
PTV 60	39.32	67.72	62.02
Brainstem	17.94	35.28	25.15
Optic chiasm	20.48	35.26	26.61
Cochlea left	1.13	13.87	2.35
Cochlea right	28.73	42.43	34.94
Lens left	2.34	4.14	2.92
Eye left	1.52	34.52	4.55
Brain	0.27	60.12	9.20
Optic nerve left	4.54	32.74	16.24
Parotid gland left	0.70	4.39	2.44
Parotid gland right	3.64	49.51	28.09
Spinal cord	1.22	23.82	8.14
Mandible	0.28	48.86	10.20

**Table 2 biomedicines-11-01538-t002:** Stereotactic radiotherapy—maximal dose analysis.

	D Max [Gy]
Brainstem	2.65
Optic chiasm	3.05
Cochlea left	0.95
Cochlea right	1.04
Lens left	0.22
Eye left	2.70
Brain (ethmoid sinus region)	9.00
Optic nerve left	2.26
Spinal cord	0.78

## Data Availability

Not applicable.

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
