# Peer review of "Stereotactic Radiotherapy Boost in Treatment of Persistent Periocular Sebaceous Carcinoma after Surgery"

_biomedicines, 2023, doi:10.3390/biomedicines11061538_

Round 1

Reviewer 1 Report

The authors describe a case of stereotactic radiotherapy boost in treatment of periocular sebaceous carcinoma.

Since residual tumor was observed at immediate post-operative imaging, the authors cannot state that it was a recurrence, but it was a tumor persistence.

Moreover, the presence of residual tumor after surgeries classifies it R2, not R1.

Please explain why chemotherapy was not administered to the patient.

Pre- and post-treatment clinical images (photos and imaging) are mandatory in a case report.

The authors should specify in the title that stereotactic radiotherapy was used for a persistence after incomplete surgery.

There are several grammatical errors. The authors should submit the text to a native English speaker to improve readability.

Reviewer 2 Report

This case report deals with the description of the treatment and the implementation of stereotactic radiotherapy for the rare case of sebaceous carcinoma. The language is adequate and the subject under investigation is relevant with the journal’s thematology, which would be of clinical interest to the readership. In general, the methodology of the study is well-structured and the clinical outcome supports the authors’ conclusions.

Prior to publication I would like the authors to kindly respond to the following questions:

The authors mention that the dose to the target was escalated up to 60 Gy with conventional fractionation. In the cited literature the dose is also escalated to a median dose of 60 Gy, with one study reaching 66.6 Gy. Could they elaborate on the rationale behind their decision to irradiate the carcinoma with a single-fraction boost of 12 Gy and why this specific dose was chosen, in spite of the outcome of the previous H&N case that was also irradiated with 12 Gy? On which radiobiological model was this based?

The GTV volume was 1.6 cm 3 . Which would be the limit of the target’s volume for which the same technique would be implemented?

Round 2

Reviewer 1 Report

Pre- and post-treatment imaging must be added in the manuscript.

Author Response

Dear reviewer,

Thank you for your comment. 
According to your suggestion, we added pre- and post-treatment imaging in the manuscript.